# Combined Metabolome and Transcriptome Analysis Revealed the Accumulation of Anthocyanins in Grape Berry (*Vitis vinifera* L.) under High-Temperature Stress

**DOI:** 10.3390/plants13172394

**Published:** 2024-08-27

**Authors:** Feifei Dou, Fesobi Olumide Phillip, Huaifeng Liu

**Affiliations:** Xinjiang Production and Construction Corps Key Laboratory of Special Fruits and Vegetables Cultivation Physiology and Germplasm Resources Utilization, Agricultural College of Shihezi University, Shihezi 832003, China; 20212312206@stu.shzu.edu.cn (F.D.); 120193202009@stu.shzu.edu.cn (F.O.P.)

**Keywords:** grape, color, peel, high temperatures, anthocyanins

## Abstract

In grape (*Vitis vinifera* L.) cultivation, high temperatures (HTs) usually reduce the accumulation of anthocyanins. In order to elucidate the regulatory mechanism of anthocyanin biosynthesis under high-temperature environments, we investigated the effects of HT stress at veraison (5% coloring of grape ears) on fruit coloration and anthocyanin biosynthesis in ‘Summer Black’ (XH) and ‘Flame seedless’ (FL) grapevines. Compared to the control group (35 °C), the total anthocyanin content of XH and FL grapes subjected to a high-temperature (HT) treatment group (40 °C) decreased significantly as the HT treatment continued, but showed an upward trend with fruit development. However, the concentration of procyanidins increased significantly following HT treatment but decreased with fruit development. Nonetheless, FL grapes showed some resistance to the HT condition, producing anthocyanin content at ripeness comparable to the control group, demonstrating a greater adaptability to HT conditions than XH grapes. Based on the CIRG index, at stage S4, the fruit of FL was classified as dark red, while XH was classified as blue-black in the control group. Anthocyanin-targeted metabonomics identified eight different types of anthocyanins accumulating in the peels of XH and FL grapes during ripening, including cyanidins, delphinidins, malvidins, pelargonidins, peonidins, petunidins, procyanidins, and flavonoids. Malvidins were the most abundant in the two grape varieties, with malvidin-3-O-glucoside being more sensitive to high temperatures. HT treatment also down-regulated the expression of structural genes and regulators involved in the anthocyanin synthesis pathways. We used the WGCNA method to identify two modules that were significantly correlated with total anthocyanin and procyanidin contents. Among them, *MYBCS1*, *bHLH137*, *WRKY65*, *WRKY75*, *MYB113-like*, *bZIP44*, and *GST3* were predicted to be involved in grape anthocyanin biosynthesis. In conclusion, this study conducted in-depth research on the HT inhibition of the biosynthesis of anthocyanins in XH and FL grapes, for reference.

## 1. Introduction

Grapes (*Vitis vinifera* L.) are among the oldest and most economically valuable berries worldwide, having been domesticated from wild plants for over 6000 years. China is the largest table grape producer and consumer, especially in Xinjiang province [1]. Color, as an important trait of grapes, not only plays an important role in quality, but also contributes to the commercial value of the fruit and its by-products. The color of grape peel is mainly determined by the anthocyanin content. Anthocyanin is a flavonoid compound that is a secondary metabolite of plants and is synthesized during maturation [2]. However, the propensity to global warming has had a severe impact on overall viticulture. Extreme temperature events are predicted to occur more frequently, intensively, and for longer periods. In several regions in Xinjiang, the daytime temperature can reach 40 °C or higher, thus affecting the normal growth of grapes, especially the accumulation of anthocyanins [3,4,5]. Over 35 °C temperature entirely obstructed anthocyanin synthesis [6].

Cyanidin, delphinidin, peonidin, petunidin and malvidin are the main anthocyanin inducers [7,8]. Phenylalanine is the precursor of anthocyanin, which is then oxidized by phenylalanine ammonia lyase (PAL), cinnamate 4-hydroxylase (C4H), and 4-coumaroyl CoA enzyme (4CL) to create 4-coumaroyl-CoA. Then, chalcone synthase (CHS) and chalcone isomerase (CHI) catalyze 4-coumaroylCoA to produce chalcone and colorless trihydroxyflavonol. Flavanone 3-hydroxylase (F3H) hydroxylates trihydroxyflavonol, forming colorless dihydroflavonol. Finally, dihydroflavonol 4-reductase (DFR) reduces colorless dihydroflavonols to leucoanthocyanidins, which are then transformed to unstable anthocyanins by anthocyanin synthase (ANS) [9,10]. To convert unstable anthocyanins into stable anthocyanins, a succession of methylation, glycosylation, and acylation processes are required. The stabilized anthocyanins are then transported from endoplasmic reticulum to the vacuoles by glutathione transferase (GST) [11,12]. Structural genes encoding *CHS*, *CHI*, *DFR*, *F3H*, *ANS*, and *3GT* involved in anthocyanin synthesis have been verified in several species [13,14,15]. In addition, transcription factors such as *MYB*, *bHLH*, *WD40*, *WRKY*, *MADS-box* and *bZIP* also participate in the synthesis of anthocyanins. MYB, bHLH, and WD40 form a complex to regulate the expression of structural genes [16,17].

These genes and transcription factors show patterns of high-temperature suppression and low-temperature enhancement [18,19,20]. Above 25 °C significantly inhibited anthocyanin accumulation in ‘Hongyang’ kiwifruit, mediated by *AcMYB1* [21]. Fang et al. [22] found that key anthocyanin structural genes, including *PAL*, *C4H*, *4CL*, *CHS*, *CHI*, *F3H*, *DFR*, *ANS*, *UFGT*, and *GST* in ‘Akihime’ plums, were more active at 20 °C than at 30 °C, with the lower anthocyanin levels under the 30 °C. However, the extreme temperature in Xinjiang can reach over 40 °C, and these studies might have limitations to explain the effect of high temperature on anthocyanins. Zhang et al. [23] studied the effect of high temperatures (38 °C and 45 °C) on the accumulation of anthocyanins in eggplant (*Solanum melongena* L.) and found that *CHSB* and *PHL11* were up-regulated and *BHLH62*, *MYB380*, *CHI3*, *CHI*, *CCOAOMT*, *AN3*, *ACT-2*, *HST*, *5MA-T1*, *CYP75A2*, *ANT17*, *RT*, *PAL2* and anthocyanin *5-aromatic acyltransferase* were down-regulated. However, most studies on the effects of high temperatures on the coloration of grape berries have been carried out in climatic chambers, where the growing conditions are strictly controlled and the temperature settings are conservative, making them more suitable for comfortable grape cultivation. The molecular mechanism of extreme high temperature (frequent high-temperature events, increased intensity, and long-lasting duration) in the Xinjiang region regulating grape coloring is poorly understood. In this study, we investigated the effect of increased temperatures on anthocyanin content and composition in ‘Summer Black’ and ‘Flame seedless’ grape berries. It also analyzed the molecular mechanism of high-temperature inhibition of anthocyanin accumulation from both metabolomic and transcriptome perspectives. The data provided will help to reveal changes in the phenotypic plasticity of the berry color of these major grape varieties under this climate change scenario. At the same time, it will deepen the understanding of the coloring of grape varieties under the influence of high temperatures, and provide new insights into the heat-tolerant breeding and cultivation management of grapes.

## 2. Materials and Methods

### 2.1. Plant Material

This experiment was conducted in 2020 in a greenhouse of the experiment station at Shihezi University, located in Shihezi, Xinjiang, China (86°4′ E, 44°18′ N). The plant materials used were ‘Summer Black’ (XH, a hybrid of Vitis vinifera × Vitis labrusca from Japan) and ‘Flame Seedless’ (FL, *Vitis vinifera* L., originating from the United States). Both varieties were grown in cultivation bags measuring 27 cm (height) × 30 cm (diameter), using a culture medium composed of pastoral soil and organic matter in a ratio of 2:1. The plant row spacing was set at 100 cm × 120 cm, and soil moisture levels were maintained between 31 and 35%. Each plot has 15 well-growing and healthy XH and FL grape plants. And three biological replicates were set up, each with 5 grape plants. The plants were pruned into a “Y” shape, with each vine having two main branches. Each of these main branches supported two berry clusters, with each fruiting branch bearing one cluster of grape bunches. Throughout the entire experimental process, all other agronomic operations were consistent.

To simulate a natural high-temperature environment, our experiments were conducted in two independent solar greenhouses. The control group (CK) was set to 35 ± 2 °C, and the high-temperature group (HT) group was set to 40 ± 2 °C. The indoor temperature was strictly monitored by temperature sensors, mainly using ventilation fans and controlling the size and ventilation time of greenhouse vents to regulate the temperature. In addition, we laid black weed-proof cloth between the rows of grape plants, effectively suppressing weed growth, and sprayed water on the ground to reduce ground temperature and promote air circulation. During periods of extreme heat, we also used shade nets to cool down and to maintain the design temperature. During the berry development stage (June to August), the average day/night temperature was 32.54/20.74 °C in the HT treatment group and 30.19/20.49 °C in the CK treatment group. The day/night temperature difference between the HT group and the CK group was (ΔT = 2.35 °C/0.24 °C) (Figure 1A). The HT period was from 10 am to 8 pm daily, and temperature management was consistent for other periods (Figure 1B). At the same time, we used the MicroLite-U disk temperature recorder to strictly monitor greenhouse temperature.

The first sample was collected at the beginning of berry coloring (about 5% coloration of the whole panicle). Then, every 10 days, samples were taken until the grape berries were completely colored. Thirty grape clusters were randomly selected for each grape variety. A total of 100 grape berries were randomly picked from the top, middle and bottom of the grape clusters, and immediately packed in an ice box and transported to the laboratory. The peel and pulp were separated. The peels were quickly frozen in liquid nitrogen and stored at −80 °C for further processing. Some fresh berries were used to determine the color index of red grape (CIRG).

### 2.2. Chromaticity Values of Grape Peel

Thirty grape berries were collected at random from each treated grape variety, with the equator serving as the boundary, and the color of the upper, middle, and lower parts of the berries was measured using a Minolta CR-200 colorimeter. These were also described by the CIE L*, a*, and b* color space coordinates [24]. The CIRG was calculated using the formula CIRG = (180 − arctangent b*/a*)/(L* + C*). Based on the CIRG values, the berry color was classified into 5 categories: green-yellow (<2), pink (2 to 4), red (4 to 5), dark red (5 to 6) and blue-black (>6.0) [25]. The L* value represented the lightness of colors, with a range of 0 to 100 (0, black; 100, white). The a* value was negative for green and positive for red. The b* value was negative for blue and positive for yellow. The chroma values (C*) = [(a*)2 + (b*)2] 0.5, hue angle, h° (0–360°) = arctangent b*/a*, and h° were on the color wheel (0° = red; 90° = yellow; 180° = green; and 270° = blue).

### 2.3. Anthocyanin and Procyanidin Content in the Peel

The total anthocyanin content in grape peel was determined using the pH difference method. Total anthocyanin and proanthocyanidin content was extracted by the Plant Anthocyanin Content Test Kit (Suzhou Keming Biotechnology Co., Ltd., Suzhou, China) and the Oligomeric Proantho Cyanidins (OPC) Kit (Suzhou Keming Biotechnology Co., Ltd., China), respectively. The UV (UV2600) spectrophotometer was used to measure the absorbance.

### 2.4. Metabolite Extraction and Data Analysis

Sample preparation and extraction: the sample was freeze-dried, ground into powder (30 Hz, 1.5 min), and stored at −80 °C until needed. A total of 50 mg powder was weighed and extracted with 0.5 mL methanol/water/hydrochloric acid (500:500:1, V/V/V). Then the extract was vortexed for 5 min, ultrasound was used for 5 min, and it was centrifuged at 12,000× *g* under 4 °C for 3 min. The residue was re-extracted by repeating the above steps again under the same conditions. The supernatants were collected, and filtrated through a membrane filter (0.22 μm, Anpel) before LC-MS/MS analysis.

UPLC Conditions: the sample extracts were analyzed using an UPLC-ESI-MS/MS system (UPLC^|^ ExionLC™ AD^|^
https://sciex.com.cn/, accessed on 12 August 2024; MS^|^Applied Biosystems 6500 Triple Quadrupole, https://sciex.com.cn/, accessed on 12 August 2024). The analytical conditions were as follows: UPLC: column, Waters ACQUITY BEH C18 (1.7 µm, 2.1 mm × 100 mm); solvent system, water (0.1% formic acid): methanol  (0.1% formic acid); gradient program, 95:5 *v*/*v* at 0 min, 50:50 *v*/*v* at 6 min, 5:95 *v*/*v* at 12 min, hold for 2 min, 95:5 *v*/*v* at 14 min; hold for 2 min; flow rate, 0.35 mL/min; temperature, 40 °C; injection volume, 2 μL.

ESI-MS/MS Conditions: linear ion trap (LIT) and triple quadrupole (QQQ) scans were acquired on a triple quadrupole linear-ion-trap mass spectrometer (QTRAP), QTRAP^®^ 6500+ LC-MS/MS System, equipped with an ESI Turbo Ion-Spray interface, operating in positive ion mode and controlled by Analyst 1.6.3 software (Sciex). The ESI source operation parameters were as follows: ion source, ESI+; source temperature 550 °C; ion spray voltage (IS) 5500 V; curtain gas (CUR) set at 35 psi. Anthocyanins were analyzed using scheduled multiple reaction monitoring (MRM). Data acquisitions were performed using Analyst 1.6.3 software (Sciex). Multiquant 3.0.3 software (Sciex) was used to quantify all metabolites. Mass spectrometer parameters including the declustering potentials (DPs) and collision energies (CEs) for individual MRM transitions were carried out with further DP and CE optimization. A specific set of MRM transitions were monitored for each period, according to the metabolites eluted within this period.

For comparison between the two groups, metabolites with fold-change (FC) ≥ 2 and fold-change (FC) ≤ 0.5 were considered as differentially accumulated metabolites (DAMs).

### 2.5. Total RNA Extraction and Transcriptome Sequencing

Total RNA was extracted from the frozen grape peel samples using the RNA prep Pure Plant Kit (Tiangen, Beijing, China), following the manufacturer’s instructions. The concentration and purity of the RNA were assessed through RNA-specific agarose gels. To enrich mRNA with polyA structures, Oligo (dT) magnetic beads were utilized, and the RNA was fragmented to about 300 bp using ion interruption. The first strand of cDNA was then synthesized with 6-base random primers and reverse transcriptase, utilizing RNA as a template, followed by the synthesis of the second cDNA strand from the first-strand template. Personalbio Biotechnology Co., Ltd. (Shanghai, China) sequenced the cDNA libraries using an Illumina sequencing platform. After creating the libraries, PCR amplification was used to enrich the library fragments. The libraries were then size-selected with a target of 450 bp. Following that, PCR products were purified with the AMPure XP system, and the libraries’ quality was assessed using the Agilent Bioanalyzer 2100 system. Differentially expressed genes (DEGs) between the two groups were analyzed using DESeq, and the *p*-value was corrected using the Benjamini and Hochberg method. Genes with *p*-value ≤ 0.05 and |log2FoldChange| > 1 were considered as DEGs. GO and KEGG function enrichment analysis was performed on DEGs, and the filter condition was *p*-value ≤ 0.05 and FDR < 0.05.

### 2.6. Validation of Gene Expression by qRT-PCR

An RNAprep Pure Plant Kit (Tiangen, Beijing, China) was used to extract the total RNA of grape peel. Reverse transcription was performed to generate cDNA using the HyperScript^TM^ III RT SuperMix for qPCR with gDNA Remover (NovaBio, Shanghai, China). For qRT-PCR experiments, the 2 × S6 Universal SYBR qPCR Mix (NovaBio, Shanghai, China) was used. This product is a SYBR Green I dye qPCR premix containing enzymes, dNTPs, buffer, fluorescent dyes, reference dyes, and other components required for qPCR. The qPCR reaction was performed in a final volume of 20 µL, consisting of 10 µL 2 × S6 Universal SYBR qPCR Mix, 2 µL of the first-strand cDNA template, 0.4 µL each of forward and reverse primers, and 7.2 µL ddH2O. The internal reference control for data normalization was Actin. The expression levels of each target gene were calculated using the 2^−ΔΔCt^ method [26]. Three biological replicates were used in all qRT-PCR experiments.

### 2.7. Data Analysis

Data were presented as the mean values of three biological replicates ± standard deviation (SD). The Origin 2021 software was used to conduct principal component analysis (PCA). Statistical analyses of the data were performed by analysis of a post hoc test (Duncan’s) and one-way factorial analysis of variance (ANOVA) using the SPSS 19.0. Significant differences were determined at a probability level of *p* ≤ 0.05. The hierarchical cluster analysis (HCA) heatmap was drawn by TBtools [27].

## 3. Results

### 3.1. High-Temperature Stress Affected the Berry Coloring and Anthocyanin and Procyanidin Contents in Grape

In terms of appearance, with the growth and development of grape berries, XH and FL grapes had poorer berry coloring compared to the control berry in the first three stages (S1, S2, S3). However, when the berry ripened (S4), there was no significant difference (Figure 2A). Based on the CIRG index, the berry of FL was classified as dark red (5 < CIRG < 6) and XH was classified as blue-black (CIRG > 6.0) at S4 (Figure 2B,C). The CIRG of FL showed a trend of first increasing and then decreasing. In the first three stages of FL grape berry development (S1 to S3), the CIRG of the control group (FCK) was higher than that of the HT treatment group (FT), and the maximum value reached at S3. Only in the S4 phase, was FCK’s CIRG lower than FT. In addition, the berry color of FCK reached dark red in S2, while the berry color of FT reached dark red in S3 (Figure 2C). The CIRG of XH grape berries showed an overall upward trend. The CIRG of the XH grape control group (XCK) was consistently higher than that of the HT treatment group (XT) of XH grapes. And the berry color of XCK reached dark red in the S3 stage, and reached blue-black in the S4 stage, while the berry color of XT reached dark red in the S4 stage (Figure 2B).

In the XH grape, the anthocyanin content continuously increased from S1 to S4. The difference in anthocyanin content between CK and T reached its maximum value in the S3 (Figure 2D). HT stress significantly reduced the anthocyanin content of FL grape from S1 to S3 of berry coloring, and reached its peak at S3, while at the S4 there was no difference between HT and CK (Figure 2E). And the content of procyanidins was continuously decreased from S1 to S4 in XH and FL grapes after HT stress. The XH grape showed significant differences in S3 (Figure 2F), while the FL grape showed significant differences in S1 and S2 (Figure 3D). There was no significant difference between the two grape varieties at S4.

### 3.2. Targeted Metabolomic Response of XH and FL Grape Peels to HT Stress

Due to significant differences in appearance color, total anthocyanin, and proanthocyanin contents observed during the coloring process under HT stress, we selected grape samples from the S3 stage to determine the anthocyanin composition, aiming to identify the substances that caused the differences between XH and FL grapes. A total of forty-three anthocyanins were identified by LC-MS/MS, including eight cyanidins, seven delphinidins, five malvidins, three pelargonidins, five peonidins, three petunidins, five procyanidins, and seven flavonoids (Appendix A). Among them, twenty-nine common anthocyanins were detected in two grape varieties, and three and seven unique anthocyanins were detected in XH and FL grapes, respectively. The HPLC profiles can be seen in Appendix A. Figure 3A shows the results of a PCA analysis to examine the sample distribution. The results showed considerable separation across all treatments, with lesser differences across the three replicates within each treatment. Principal component 1 (PC1) efficiently distinguished between the FL and XH grape types, while principal component 2 (PC2) distinguished XCK from XT. These findings suggested significant changes in the anthocyanin components of the two grape varieties when exposed to high-temperature (HT) stress. As shown in Figure 3B, HT stress significantly reduced the anthocyanin contents of XH grape, while there was no significant difference in the anthocyanin content of FL grape. In the XH grape, the anthocyanin contents were mainly composed of malvidins, peonidins and procyanidins. After HT treatment, the contents of malvidins and peonidins significantly decreased by 1246.06 and 326.18 µ/kg, respectively. The FL grape’s anthocyanin concentration was mainly composed of malvidins, peonidins, and cyanidins. Although HT had a lower effect on anthocyanins in FL compared to XH, malvidin still significantly decreased, by 348.58 µ/kg. The other key anthocyanin contents can be viewed in Appendix A. Further KEGG analysis showed that these DAMs were enriched in the “Metabolic”, “Isoflavonoid biosynthesis”, “Flavonoid biosynthesis”, “Flavone and flavonol biosynthesis”, “Biosynthesis of secondary metabolites” and “Anthocyanin biosynthesis” pathways (Figure 3C).

Hierarchical clustering analysis (HCA) showed that HT had a greater impact on the anthocyanins in XH than in FL (Figure 3D,E). A total of 30 anthocyanins decreased in XH after HT stress, while only 6 anthocyanins decreased in FL. Orthogonal partial least squares discriminant analysis (OPLS-DA) is an analytical method that visualizes data and quantifies the degree of differences between samples through the correlation among the data. Screening of the DAMs between normal temperature and HT was based on the variable importance in projection (VIP) value greater than 1 and *p*-value < 0.05. Seven and one DAMs were screened in XH and FL, respectively (Figure 3F,G). The DAMs in XH were malvidin-3-O-glucoside, malvidin-3,5-O-diglucoside, peonidin-3,5-O-diglucoside, petunidin-3-O-glucoside, peonidin-3-O-glucoside, delphinidin-3-O-glucoside, and cyanidin-3-O-(6-O-p-coumaroyl)-glucoside, while in FL it was Malvidin-3-O-glucoside.

### 3.3. Transcriptional Responses to HT Stress of XH and FL Grapes

In order to further understand the impact of high temperature on anthocyanin biosynthesis during grape berry development, we conducted RNA sequencing and DEG analysis on the entire berry-color transition period (S1 to S4). In our study, RNA libraries of 48 grapes (XH and FL) were prepared and sequenced (Appendix A). As shown in Appendix A, the percentage of clean reads and clean data in each library varies from 90.66% to 92.42%, with Q30 values more than 90%. This showed that the basic detection accuracy for each read was more than 99.9%. PCA analysis was performed to give an overview of all HT-treated grape samples (Appendix A). The XH and FL grapes were easily identified, indicating significant differences in their transcriptome. Furthermore, significant differences were observed in XH transcriptome following HT.

### 3.4. Analysis of DEGs of XH and FL Grapes

We conducted comparative analysis on differentially expressed genes (DEGs, *p*-value ≤ 0.05 and |log2 fold change| ≥ 1) among grape varieties exposed to different temperature treatments. We observed a significant increase in DEGs in XH and FL grapes with the growth and development of the berries, and, at different stages, HT stress led to more gene down-regulation rather than up-regulation (Figure 4A,B). Meanwhile, in the control and high-temperature treatment groups, we identified 1600 and 554 DEGs shared among the XH and FL grapes (Figure 4C,D), respectively. And 30 and 38 core DEGs were shared at different stages among the XH and FL grapes, respectively (Figure 4E,F).

### 3.5. Functional Enrichment Analysis of DEGs of XH and FL Grapes

The GO enrichment analysis of DEGs observed in XCK vs. XT and FCK vs. FT (Appendix A) revealed that DEGs in XCK vs. XT were primarily distributed in biological processes and cellular components, whereas DEGs in FCK vs. FT were primarily distributed in biological processes and molecular functions. The physiological processes involved 12,297 DEGs in XCK vs. XT and 10,878 DEGs in FCK vs. FT. As a result, HT treatments influenced the expression of genes associated with physiological functions.

To elucidate the biochemical metabolism or signal transduction pathways in which various genes may participate in different samples, KEGG pathway enrichment analysis was performed. The results showed that DEGs in response to HT of XH and FL grape peels displayed significant enrichment in terms related to “Phenylpropanoid biosynthesis”, “Flavonoid biosynthesis”, “Plant hormone signal transduction”, and “Protein processing in endoplasmic reticulum” (Appendix A). These pathways were mainly the metabolic pathway, genetic information processing and environmental information processing. These results showed that these metabolic processes played an important role in the response to HT stress.

To gain a better understanding of the roles of DEGs, we visualized the DEGs using MapMan software (Appendix A). The MapMan results revealed that DEGs in XCK vs. XT and FCK vs. FT were primarily involved in metabolic pathways, particularly secondary metabolism, protein homeostasis, lipid metabolism, phytohormone action, redox homeostasis, carbohydrate metabolism, and photosynthesis, with the majority of genes involved in these pathways being down-regulated.

### 3.6. Transcription Family (TF) Genes Respond to HT Stress of XH and FL Grapes

We observed that transcripts encoding genes were categorized into various families, including bHLHs, ERFs, NACs, MYBs, WRKYs and C2H2s (Figure 5A). There were 1708 shared genes differentially expressed in two grapes after HT stress, with 1713 and 2342 genes differentially expressed in XH and FL grapes, respectively (Figure 5B). Using KEGG annotation, these genes were majorly related to environmental information processing (plant hormone signal transduction, etc.), metabolism (phenylpropanoid biosynthesis, flavonoid biosynthesis, photosynthesis—antenna proteins, etc.), organismal systems (circadian rhythm, plant, etc.) and genetic information processing (protein processing in endoplasmic reticulum, etc.) (Figure 5C). We used the k-means clustering algorithm to divide 1708 DEGs into six clusters according to their expression patterns (Figure 5D). By analyzing these six clusters, we found that genes expression in class 1, class 2 and class 4 showed an upward trend in berry development. The expression patterns of class 3 and class 6 genes showed an upward–downward trend, but class 3 was classified into more genes. However, the genes in class 5 showed a downward trend in berry development. It is worth noting that a large number of heat shock protein genes and transcription factor (TFs) genes were classified as class 1 and class 2. HT stress led to significant up-regulation of these genes during berry development. In summary, these TF genes were potential key regulators that may participate in regulating grape-berry HT adaptation.

### 3.7. Key DEGs and DAMs Associated with Production of Anthocyanins in XH and FL Grapes after HT

Figure 6 clearly shows the differences in gene transcription levels throughout the anthocyanin and flavonoid biosynthesis pathways in XH and FL grape berries, following HT treatment. There were 72 enzyme genes identified, including *4CL* (3), *ANS* (1), *BZL* (1), *C3′H* (1), *C4H* (3), *CAD* (11), *CCR* (1), *CHS* (5), *F3′5′H* (1), *F3H* (1), *F5H* (2), *FLS* (3), *LAR* (2), *PAL* (13), and *POD* (24). These large transcripts of structural genes were essential for grape anthocyanin and flavonoid synthesis. In lignin metabolism, a large number of *POD*, *CAD* and *CCR* genes were up-regulated in XH grapes and down-regulated in FL grapes after HT treatment. In the phenylpropanoid pathway, flavonoid biosynthesis and flavonoid and flavonol biosynthesis, HT treatment down-regulated the expression of *PAL*, *C4H*, *CHS*, *F3H*, *FLS*, *LAR*, *BZ1*, and *ANS* genes. Twelve key anthocyanin synthesis genes were selected for qRT-PCR analysis to validate the RNA-Seq data (Appendix A). The expression patterns of qRT-PCR and RNA-seq data were highly consistent. Therefore, these genes may affect the accumulation of anthocyanins in grape berries by responding to high-temperature stress. The DAMs of flavonoids were located downstream of the phenylpropanoid pathway, and there was a significant difference in the accumulation of dihydrokaempferol content in XH and FL berry peels. In flavone and flavonol biosynthesis, we detected differential accumulation of nictoflorin and rutin products in XH and FL grapes after HT treatment. Meanwhile, FL grapes accumulated a large amount of DAMs (Cyanidin 3-O-glucoside, Peonidin 3-O-glucoside, Delphinidin 3-O-glucoside, Malvidin 3-O-glucoside, Petunidin 3-O-glucoside) compared to XH grapes, while the accumulation of these DAMs in XH was significantly reduced, consistent with the trend observed in gene expression. Generally, under HT conditions, a large number of genes involved in the anthocyanin pathway were down-regulated, and anthocyanin metabolism in grape peel was inhibited.

### 3.8. Anthocyanin, Procyanidin and DEG Co-Expression Networks during Grape Peel Development

In order to further explore the relationship between transcriptome with anthocyanin and procyanidin content in grape peel, we constructed grape-peel weighted gene co-expression network analysis (WGCNA), and analyzed the key DEG network related to anthocyanin biosynthesis. Correlating co-expression networks with traits, a total of 11 WGCNA modules were identified (Figure 7A). Among them, the “MEblue” and “MEmagenta” modules had the highest correlation with the total anthocyanin content, while the “red” and “turquoise” modules had the highest correlation with the procyanidin content. KEGG enrichment analysis of the DEGs found in these modules showed that these pathways were mainly metabolic pathways and environmental information processing (Appendix A). And DEGs in the module were mainly involved in “Flavonoid biosynthesis”, “Protein processing in endoplasmic reticulum”, “Plant hormone signal transduction”, and “Photosynthesis—antenna proteins and photosynthesis”. Nine genes in the “blue” and “turquoise” modules were anthocyanin structural genes: one *CHS*, two *4CLs*, one *LAR*, one *FLS5*, one *F3H*, one *5,3GT*, one *UGT88F3* and one *CYP81E*. The other genes were key transcription factors: three *MYBs*, one *bZIP*, one *bHLH*, and two *WRKYs*. In addition, there were also some key genes: five *GSTs*, one *AAO* and three *PODs* in these modules (Figure 7B,C). Compared with the control treatment, most of the genes showed high expression in the S1 and S2 stages of HT treatment, while S3 and S4 showed a low expression in the HT treatment (Figure 7D,E). The gene members of the gene correlation network are shown in Appendix A.

## 4. Discussion

‘Summer Black’ and ‘Flame Seedless’ grapes were chosen for this study because they were the main table grape varieties in Xinjiang and have different anthocyanin compositions and agronomic properties. ‘Flame Seedless’, a popular table grape type grown by the Xinjiang Production and Construction Corps, produced bright red or purple-red berries of high quality [28,29]. In contrast, when ripened, ‘Summer Black’ grapes had a purple-black or blue-black hue, were easy to color, and had a high tolerance to stress and adaptability to cultivation [30]. High temperatures, an important abiotic stress factor, considerably impede the production of anthocyanins, hence reducing the quality of grape berries [31]. In our previous research, we found that ‘Flame seedless’ had stronger heat resistance than ‘Summer Black’ [32], which meant that they might have differences in anthocyanin accumulation under high-temperature stress. Therefore, we simulated the natural high-temperature environment under solar greenhouse conditions to study the effects of high temperature on anthocyanin content and composition of two major red grape varieties. The results showed that high-temperature treatment had a significant inhibitory effect on berry peel coloration, and delayed the coloring of ‘Summer Black’ and ‘Flame Seedless’ grapes during veraison. High-temperature stress caused ‘Summer Black’ to not achieve blue-black even at the mature stage, whereas ‘Flame Seedless’ could achieve purple-red. Anthocyanin accumulation differed significantly between high and normal temperatures in ‘Summer Black’ grapes, but not in ‘Flame Seedless’ berries, throughout ripening. Thus, while the anthocyanin concentration in ‘Flame Seedless’ initially reduced during the middle of the coloring process due to high temperatures, the variety could mitigate the effects of the high-temperature environment, to some degree. During the mature-berry stages, ‘Flame Seedless’ developed anthocyanin levels comparable to those of control berries. Procyanidins were the precursor for anthocyanin synthesis, and a high temperature inhibited the conversion of procyanidins to anthocyanins, resulting in higher levels of procyanidins at 40 °C than at 35 °C, in our study. Reduced total anthocyanin content due to a high temperature was also observed in other grape varieties [19,33].

Anthocyanins were the main pigments responsible for the red, purple and blue colors in plants, and grape peel coloration was the result of anthocyanin accumulation [34]. In grapes, the main anthocyanins were malvidins, petunidins, delphinidins, flavonoids, and peonidins [35]. Among them, cyanidins and peonidins were the main pigments that appeared red, delphinidin was the main pigment that appeared blue, malvidin was the main pigment that appeared purplish red, and petunidin was the main pigment that appeared purple [36]. In this study, a total of 43 anthocyanins were detected in ‘Summer Black’ and ‘Flame Seedless’ grape peels, which could be classified as cyanidins, delphinidins, malvidins, pelargonidins, peonidins, petunidins, procyanidins, and flavonoids. In ‘Summer Black’ grape, the mainly anthocyanins were malvidins, peonidins and procyanidins, and high-temperature stress significantly reduced the malvidin and peonidin contents, which might be the main reason why ‘Summer Black’ grapes did not reach blue-black at the mature stage. Meanwhile, the malvidin content was significantly reduced in ‘Flame Seedless’ grapes under high-temperature stress. It indicated that high temperature reduced the malvidin content, further affecting the coloring of red grapes. Ju et al. [37] also found that malvidins were correlated with the red color of the peel and flesh of the ‘Yan73’ grape. OPLS-DA was a supervised dimension-reduction method, which could identify complex variables, and evaluate the differences and regularity between samples [38]. We screened seven DAMs (malvidin-3-O-glucoside, malvidin-3,5-O-diglucoside, peonidin-3,5-O-diglucoside, petunidin-3-O-glucoside, peonidin-3-O-glucoside, delphinidin-3-O-glucoside, and cyanidin-3-O-(6-O-p-coumaroyl)-glucoside) in ‘Summer Black’; only malvidin-3-O-glucoside was screened in ‘Flame Seedless’, which was significantly reduced under high-temperature stress.

There were significant differences in the expression characteristics of anthocyanin-synthesis structural genes and related regulatory genes among different crop species or different varieties of the same crop [39]. By analyzing the biosynthetic pathways of anthocyanins and flavonoids, a total of 72 differentially expressed key structural genes were identified, including *4CL*, *ANS*, *BZL*, *C3′H*, *C4H*, *CAD*, *CCR*, *CHS*, *F3′5′H*, *F3H*, *F5H*, *FLS*, *LAR*, *PAL* and *POD*. Therefore, the abundance of structural gene transcripts was an important condition for the accumulation of anthocyanins and flavonoids in grapes. It was worth noting that a large number of genes involved in the anthocyanin pathway were down-regulated under high-temperature conditions, the anthocyanin metabolism in grape peel was inhibited, and there were significant differences in the expression characteristics of these genes in ‘Summer Black’ and ‘Flame Seedless’ grape peels.

Previous studies used WGCNA analysis to identify the key genes that lead to phenotypic differences [40], and provided a reference for this study to explore the regulatory genes and modules of anthocyanin synthesis in grape peel. Using the WGCNA method, a weighted gene co-expression network was constructed based on transcriptome sequencing data at different developmental stages of grape peel, and the modules significantly related to the content of total anthocyanins and procyanidins were identified: “MEblue”, “MEmegenta”, “MEred” and “MEturquoise”. In these modules, eight anthocyanin structural genes were identified, including one *CHS*, one *F3H,* two *4CLs*, one *FLS5*, one *LAR*, one *5,3GT*, and one *UGT88F3*. Compared with the control treated grape, the expression of these anthocyanin structure genes in high-temperature-treatment grapes was significantly lower in the development process of grapes, indicating that they played a key role in the inhibition of anthocyanin biosynthesis by high temperature. These observations were similar to those of previous studies [41,42].

In the study of anthocyanin-synthesis regulatory genes, many types of transcription factors were involved in regulating anthocyanin biosynthesis, such as the well-known *ERFs*, *bHLHs*, *NACs*, *bZIP*, *MYBs* and *WRKYs*. These TFs were differentially expressed in the grape peel during different stages of development, and might be involved in regulating the adaptation of the grape berry to high temperature. In the “MEblue” and “MEturquoise”, we find different TFs, including *MYBs*, *bHLHs*, *WRKYs*, *bZIPs* and *ERFs*, some of which had been shown to regulate anthocyanin synthesis. For example, *MybA1* and *MYBCS1*, members of the MYB super-family, could catalyze the reaction from the colorless leucoanthocyanidins to colored anthocyanins, thereby regulating the accumulation of anthocyanins in the peel of Muscadine berry peels [43]. *SmMYB113* over-expression promoted anthocyanin accumulation in eggplant peels and pulps [44]. In the “MEblue”, *MYB113-like* was significantly differentially expressed after high-temperature treatment, and was highly positively correlated with anthocyanin synthesis. MdWRKY75 bound to the promoter of MdMYB1 and promoted the accumulation of anthocyanins in the callus of apple ‘Orin’ [45]. And in the pears, PbWRKY75 promoted anthocyanin synthesis by activating the expression of *PbDFR*, *PbUFGT*, and *PbMYB10b* genes [46]. Phytohormone abscisic acid (ABA)-induced apple MdbZIP44 enhanced the binding of MdMYB1 to downstream target-gene promoters and promoted the accumulation of anthocyanins [47]. Most importantly, in our study, *MYBCS1*, *bHLH137*, *WRKY65*, and *WRKY75* identified in “MEturquoise” were significantly negatively correlated with anthocyanin content, while positively correlated with procyanidin content. In addition, these TFs were significantly down-regulated after high-temperature treatment, suggesting that they might be involved in regulating the inhibitory effect of high temperature on grape peel anthocyanin accumulation.

In addition to accelerating pigment degradation, high temperature also inhibited the rate of anthocyanin biosynthesis in early maturation, and the gene expression and enzyme activity were lower activities [19,35]. The results of the study showed that high temperature increased the degradation of anthocyanins, which may be due to the fact that high temperature can stimulate peroxidase activity, which degraded the anthocyanins in grape berries [48,49]. In this study, a large number of peroxidase genes were up-regulated before ripening and down-regulated after ripening, especially in the ‘Flame Seedless’ grape, which might be related to the heat tolerance of ‘Summer Black’ and ‘Flame Seedless’ grapes. The ‘Flame Seedless’ was more adaptable to high temperature. Anthocyanins were synthesized on the cytoplasmic surface of the endoplasmic reticulum (ER) and then transported to vacuoles for storage, where glutathione S-transferases (GSTs) were involved. The important role of GSTs in anthocyanin accumulation has been reported in pear [50], Chinese bayberry [51], grape [52], cotton [53], and tree peony [54]. In this study, the expression of *GST3* (*VIT_05s0049g01090*) and *GST-like* (*VIT_07s0005g04890*) were up-regulated with the development of ‘Summer Black’ and ‘Flame Seedless’ grape berries under high-temperature treatment. These genes were candidate participants that were predicted to facilitate anthocyanin transport and promote pigment accumulation. Therefore, it was likely that these two processes (pigment degradation and anthocyanin biosynthesis) occurred in both ‘Summer Black’ and ‘Flame Seedless’ grape peels in high-temperature conditions and could explain our results.

## 5. Conclusions

In this study, a total of 43 anthocyanins accumulated in the peel during ripening. Among them, we screened seven components and one anthocyanin component in ‘Summer Black’ and ‘Flame Seedless’, respectively, and found a significant decrease in their content, which was related to the color change of the grape peel. Transcriptome analysis showed that a high temperature led to the down-regulation of genes related to anthocyanin synthesis (*4CL*, *ANS*, *BZL*, *C3′H*, *C4H*, *CAD*, *CCR*, *CHS*, *F3′5′H*, *F3H*, *F5H*, *FLS*, and *LAR*) and the up regulation of genes related to peroxidase (*PAL* and *POD),* thereby increasing anthocyanin degradation. Some transcription factors, MYBCS1, bHLH137, WRKY65, WRKY75, MYB113-like, bZIP44, and GST3 were predicted to be involved in grape anthocyanin biosynthesis, but further research was needed. Overall, the results of this study suggested that heat stress corresponding to the predicted local extreme high temperatures reduced anthocyanin levels in ‘Flame Seedless’ and ‘Summer Black’ berries. The ‘Flame Seedless’ showed greater resilience and was more suitable for planting in high-temperature areas and as a breeding parent for heat-tolerant grapes. Meanwhile, the genes discovered in this study provided useful insights into the molecular mechanism of high-temperature inhibition of grape anthocyanin accumulation. However, the regulatory patterns between these transcription factors and genes were still unclear, and require further research for validation.

## Figures and Tables

**Figure 1 plants-13-02394-f001:**
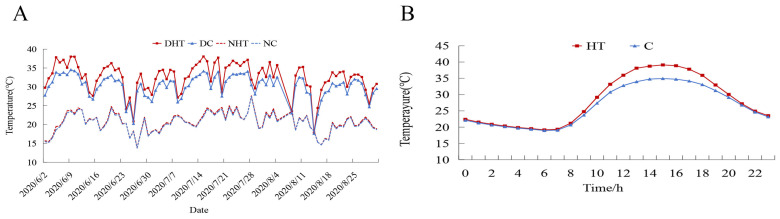
(**A**) Temperature changes during fruit development from June to August 2020. (**B**) Average temperature (°C) of high-temperature (HT) and control treatments (C). DHT: daily average temperature of high-temperature treatment; NHT: night average temperature of high-temperature treatment; DC: daily average temperature of control treatment; NC: night average temperature of control treatment.

**Figure 2 plants-13-02394-f002:**
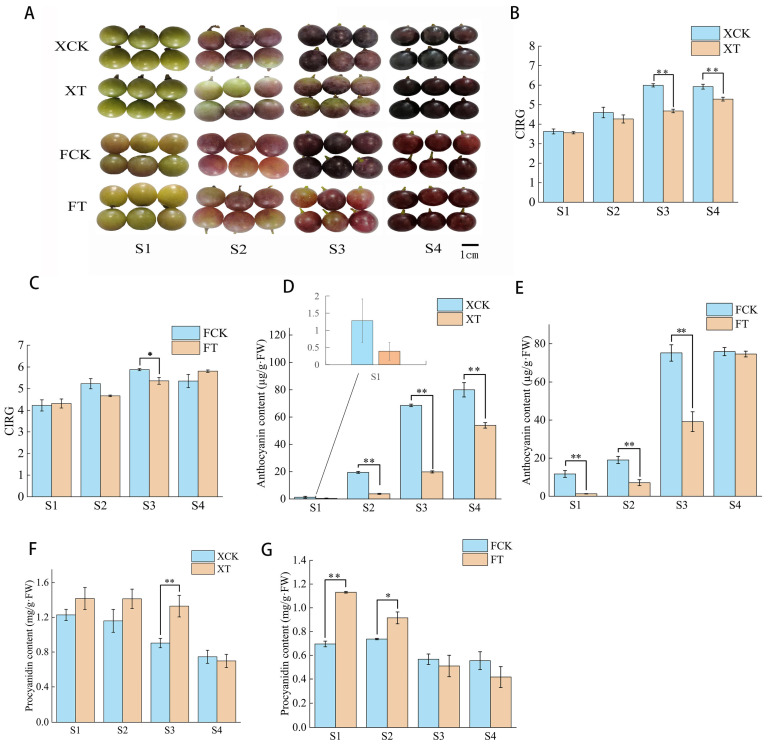
High-temperature treatment affected the color phenotype of XH and FL grape berries. The appearance (**A**) of grape berries, and CIRG of XH (**B**) and FL (**C**) grapes, total anthocyanin contents of XH (**D**) and FL (**E**) grape, and procyanidin contents of XH (**F**) and FL (**G**) grape. FW: fresh weight. S1, S2, S3, and S4 represent 0, 10, 20, 30, and 40 days after 5% coloration of whole panicle, respectively. Values stand for means ± standard deviation (SD) of three independent biological replicates. The vertical bars indicate the SDs. Statistical significance was measured using Student’s *t*-test (*: *p* < 0.05, **: *p* < 0.01). This is the same in the following.

**Figure 3 plants-13-02394-f003:**
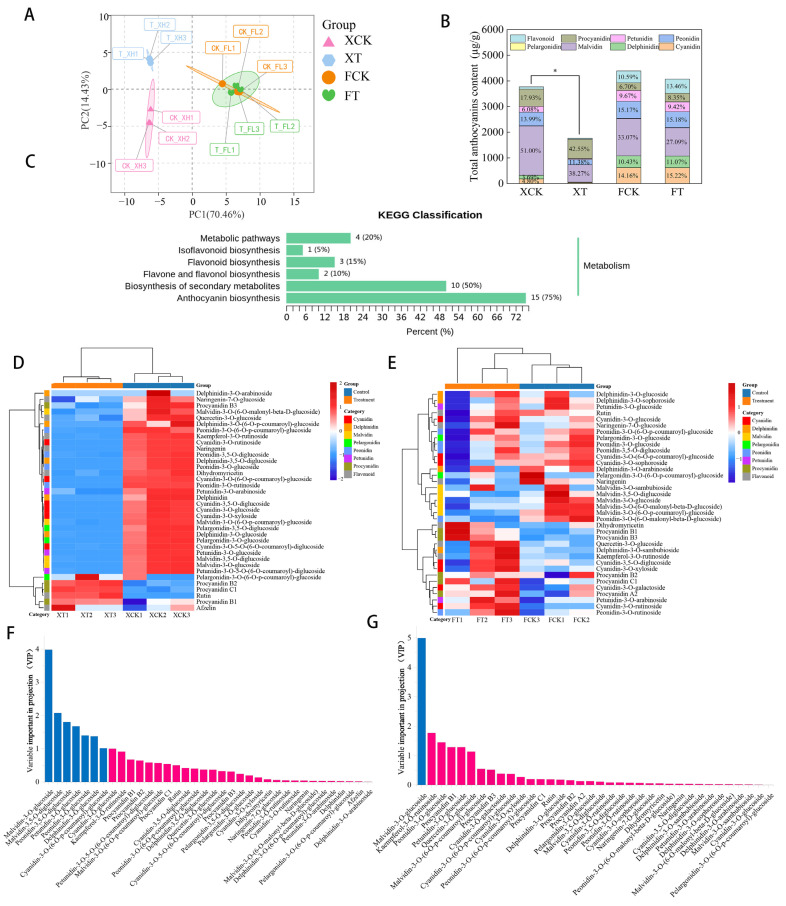
Targeted metabolomic analysis of XH and FL grape peels after HT treatment. (**A**) PCA of metabolites from HT treatment of XH and FL. (**B**) Stacked bar chart of relative changes in metabolite content. Statistical significance was measured using Student’s *t*-test (*: *p* < 0.05). The *x*-axis and *y*-axis represent different treatments and anthocyanin contents, respectively. (**C**) KEGG pathway enrichment analysis of different metabolites. (**D**) The cluster heat map of anthocyanins in XH grape peels under HT treatments. (**E**) The cluster heat map of anthocyanins in FL grape peels under HT treatments. (**F**) The variable important in projection of OPLS-DA based on the anthocyanins detected in XH. (**G**) The variable important in projection of OPLS-DA based on the anthocyanins detected in FL. The blue column represents DAMs with VIP > 1 and *p*-value < 0.05, the pink column represents DAMs with VIP < 1 and *p*-value > 0.05.

**Figure 4 plants-13-02394-f004:**
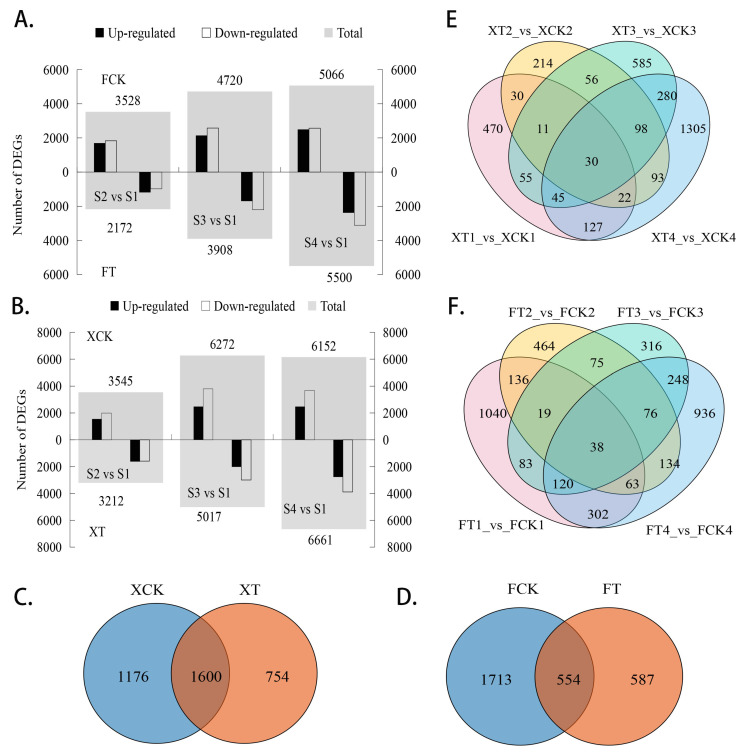
DEG analysis in XH and FL grape peels under HT treatment. (**A**) The quantity statistics of DEGs in different stages of XCK and XT. (**B**) The quantity statistics of DEGs in different stages of FCK and FT. (**C**) Venn diagram depicting the shared DEGs between XCK and XT. (**D**) Venn diagram depicting the shared DEGs between FCK and FT. (**E**) Venn diagram depicting the shared core and specific DEGs of XH grape among four color-transition periods. (**F**) Venn diagram depicting the shared core and specific DEGs of the XH grape among four color-transition periods. XCK1, XCK2, XCK3, and XCK4 represent the S1, S2, S3, and S4 of the XH grape in the control group, respectively; XT1, XT2, XT3, and XT4 represent the S1, S2, S3, and S4 of the XH grape in the HT treatment group, respectively; FCK1, FCK2, FCK3, and FCK4 represent the S1, S2, S3, and S4 of the FL grape in the control group, respectively; FT1, FT2, FT3, and FT4 represent the S1, S2, S3, and S4 of the FL grape in the HT treatment group, respectively. This is the same in the following.

**Figure 5 plants-13-02394-f005:**
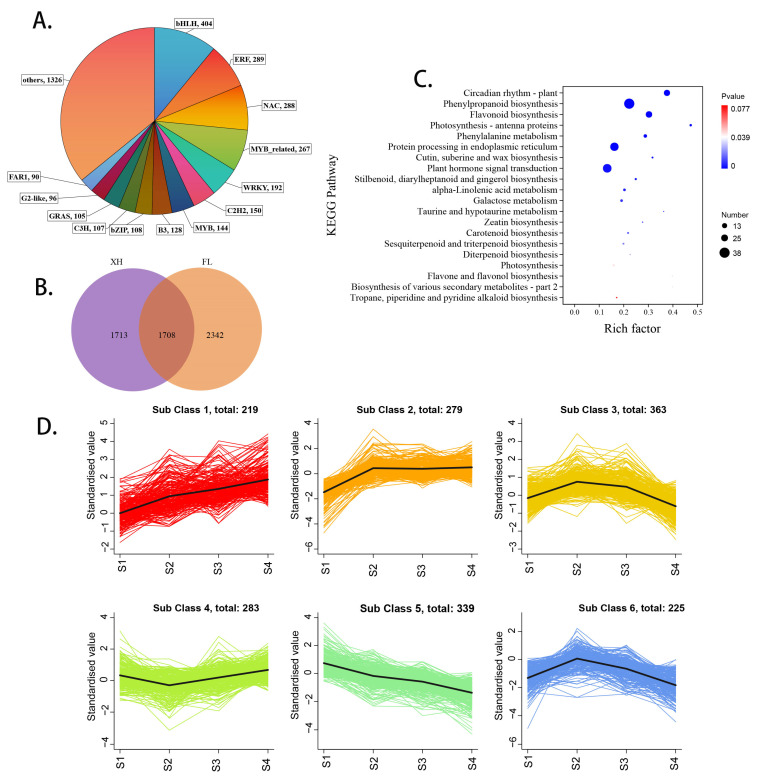
Analysis of transcription families in grape fruit under HT conditions. (**A**) Main differentially expressed transcription families. (**B**) Venn diagram shows the number of genes between XH and FL grapes. (**C**) KEGG analysis of shared genes in the two grapes. (**D**) The k-means clustering divided the DEGs profiles of XH and FL berries at different developmental stages into six clusters. The *x*-axis represents the four stages of fruit growth and development in grapes, while the *y*-axis represents the log2fc value of TF differential expression.

**Figure 6 plants-13-02394-f006:**
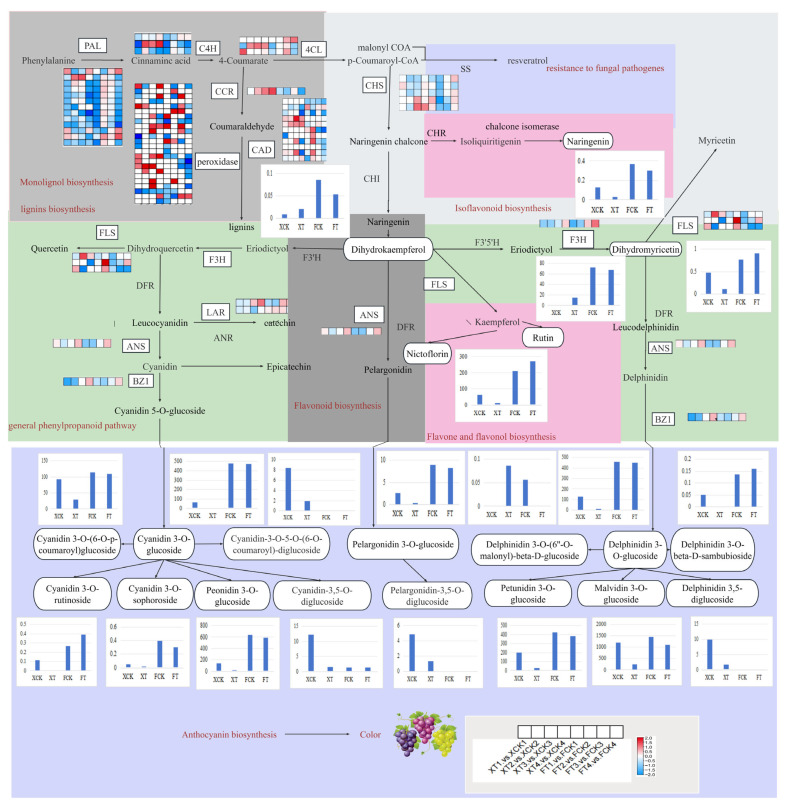
Expression levels of structural DEGs and DAMs involved in anthocyanin biosynthesis pathways in XH and FL grapes after HT treatment. The white filled box represents DEGs, and the heatmap represents the expression level of DEGs of XH and FL. Red indicates a significant up-regulation of DEGs, while blue indicates a significant down-regulation of DEGs. The white filled ellipse represents the identified DAMs in the metabolome, and the bar graph represents the accumulation level of DAMs of XH and FL. PAL, phenylalanine ammonia lyase gene; C4H, trans-cinnamate 4-monooxygenase gene; 4CL, 4-coumarate–CoA ligase gene; CHS, chalcone synthase gene; F3H, naringenin 3-dioxygenase gene; ANS, anthocyanidin synthase gene; BZ1, anthocyanidin 3-O-glucosyltransferase gene; F3′5′H, flavonoid 3′,5′-hydroxylase gene; F3′H, flavonoid 3′-monooxygenase gene; DFR, bifunctional dihydroflavonol 4-reductase/flavanone 4-reductase gene; CHI, chalcone isomerase gene; FLS, flavonol synthase gene; LAR, leucoanthocyanidin reductase gene; CCR, cinnamoyl-CoA reductase gene; CAD, cinnamyl-alcohol dehydrogenase gene.

**Figure 7 plants-13-02394-f007:**
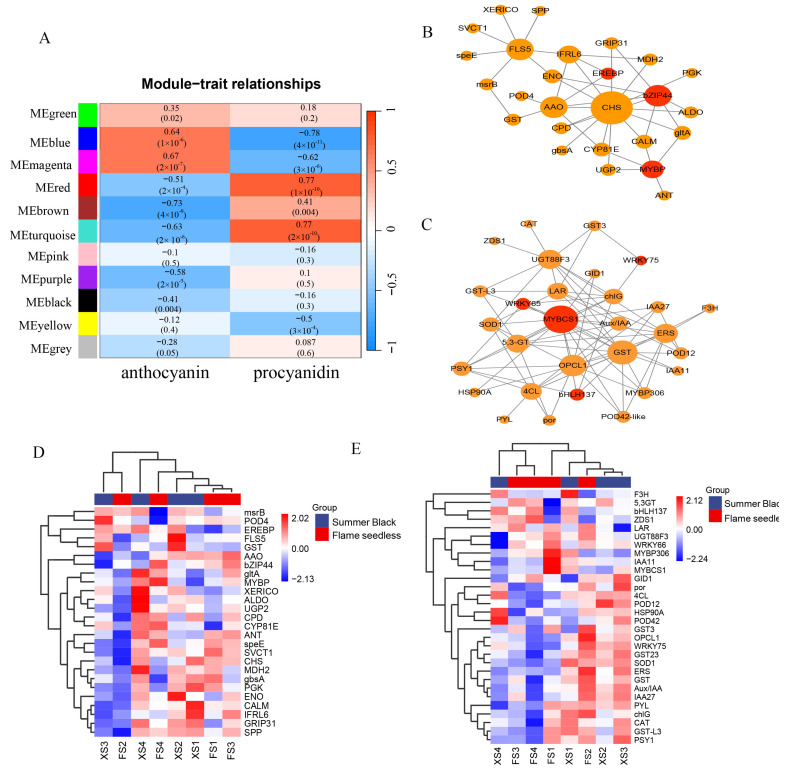
Co-expression network analysis of genes in modules at control temperature and HT treatment in two grape varieties. (**A**) Modules most associated with anthocyanins and procyanidin. The left panel shows 11 modules and the color scale on the right shows the correlation between −1 and 1. (**B**) Connectivity network between 26 key anthocyanin-related genes in the “blue” module. (**C**) Connectivity network between 31 key anthocyanin-related genes in the “turquoise” module. (**D**) Heatmap of 26 genes expression in the “blue” module. (**E**) Heatmap of 31 genes’ expression in the “turquoise” module. The values of the blue-to-red gradient bars indicate the log2-fold change relative to the control sample. XS1, XS2, XS3 and XS4 represent S1, S2, S3, and S4 of the XH grape, respectively. FS1, FS2, FS3 and FS4 represent S1, S2, S3, and S4 of the FL grape, respectively.

## Data Availability

Data are contained within the article.

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
