# Peer review of "Combined Metabolome and Transcriptome Analysis Revealed the Accumulation of Anthocyanins in Grape Berry (Vitis vinifera L.) under High-Temperature Stress"

_plants, 2024, doi:10.3390/plants13172394_

Round 1

Reviewer 1 Report

Comments and Suggestions for Authors

1.     The manuscript includes KEGG/GO enrichment analysis; however, the results section does not report the significance enrichment outcomes, and the materials and methods section lacked details on the FDR correction. Without significant results, can the findings be considered reliable?

2.     In the caption of Fig. 4, clarifying the threshold values for significance (P-value and |log2 fold change| in 305 line) would be helpful.

3.     In lines 42-45 mentioned, “In several regions in Xinjiang, the daytime temperature can reach 40 °C or higher, thus affecting the normal growth of grapes, especially the accumulation of anthocyanins. Over 35 °C temperature entirely obstructed anthocyanin synthesis.” For the temperature scale, the manuscript compares the control group (35°C) and the high-temperature (HT) group (40°C). Is selecting a critical temperature threshold of 35°C appropriate in this context? Consider selecting multiple temperature thresholds or scales to ensure that anthocyanin synthesis and content under normal conditions are accurately established. This will provide a reliable basis for comparing the reduction process observed under high-temperature conditions.

Author Response

Point 1: The manuscript includes KEGG/GO enrichment analysis; however, the results section does not report the significance enrichment outcomes, and the materials and methods section lacked details on the FDR correction.Without significant results, can the findings be considered reliable?

Response 1: We have added these contents to the manuscript.

3.5 Functional enrichment analysis of DEGs of XH and FL grapes

We conducted KEGG/GO enrichment analysis. We found that HT treatments main influenced the expression of genes associated with physiological functions. And the KEGG results showed that DEGs in response to HT of XH and FL grape peels displayed significant enrichment in terms related to "Phenylpropanoid biosynthesis", "Flavonoid biosynthesis", "Plant hormone signal transduction", and "Protein processing in endoplasmic reticulum" .

2.5 Total RNA extraction and transcriptome sequencing

Genes with P-value ≤ 0.05 and |log2FoldChange| > 1 were considered as DEGs. GO and KEGG function enrichment analysis was performed on DEGs, and the filter condition was P-value ≤ 0.05 and FDR < 0.05.

Point 2: In the caption of Fig.4, clarifying the threshold values for significance(P-value and|log2 fold change|in 305 line)would be helpful.

Response 2: We clarified the threshold values for significance in the manuscript.

3.4 Analysis of DEGs of XH and FL grapes

We conducted comparative analysis on differentially expressed genes (DEGs, P-value ≤ 0.05 and |log2 fold change| ≥ 1) among grape varieties exposed to different temperature treatments.

Point 3: In lines 42-45 mentioned, “In several regions in Xinjiang,the daytime temperature can reach 40°C or higher,thus affecting the normal growth of grapes, especially the accumulation of anthocyanins. Over 35°C temperature entirely obstructed anthocyanin synthesis.” For the temperature scale, the manuscript compares the control group (35°C) and the high-temperature (HT) group (40°C). Is selecting a critical temperature threshold of 35°C appropriate in this context? Consider selecting multiple temperature thresholds or scales to ensure that anthocyanin synthesis and content under normal conditions are accurately established. This will provide a reliable basis for comparing the reduction process observed under high-temperature conditions.

Response 3: In general, grapes grown in cooler climates appear to be more sensitive to temperature, while other grapes grown in hotter climates appear to be able to maintain secondary metabolism in more extreme high temperatures (HT). Grape varieties have a strong dependence on HT, and the optimal temperature of plants grown in HT environment for a long time can change with the change of ambient temperature, while the short-term HT stress response is slow or very mild, which is not enough to assess the actual adaptability of plants to HT stress. Some variety (such as ‘Cabernet Sauvignon’ grapevine) showed no damage to photosynthesis and growth at 35℃, and even improved compared to 25℃ conditions.Thus, although the anthocyanin content initially decreases due to HT (when half-ripe), heat-tolerant varieties can override the effects of HT conditions to some extent and produce anthocyanin content comparable to that of control berries at harvest. For example, even if just attempting to compare effects of day and night temperatures, means and/or maxima ranging from 20 ℃ to 45 ℃ have been applied at night and from 30℃ to 49 °C during the day. On the other hand, some grape berries grown at hot-day (35℃) and warm-night (25 or 30℃) had a higher level of anthocyanins than fruit ripened at hot-day cool-night (15 or 10℃). In order to better adapt the actual production environment of the grapes, we chose 35℃ as the control, which may still be higher than other regions, but it is a common temperature for Xinjiang. And 40℃ as the treatment group, which is more in line with the extreme heat in Xinjiang. Better evaluation of the actual adaptability of grapes to high temperature stress.

Reference

de Rosas, I., Ponce, M. T., Malovini, E., Deis, L., Cavagnaro, B., & Cavagnaro, P. (2017). Loss of anthocyanins and modification of the anthocyanin profiles in grape berries of Malbec and Bonarda grown under high temperature conditions. Plant science : an international journal of experimental plant biology, 258, 137–145. https://doi.org/10.1016/j.plantsci.2017.01.015

Yamori, W., Hikosaka, K., & Way, D. A. (2014). Temperature response of photosynthesis in C3, C4, and CAM plants: temperature acclimation and temperature adaptation. Photosynthesis research, 119(1-2), 101–117. https://doi.org/10.1007/s11120-013-9874-6

Gouot, J. C., Smith, J. P., Holzapfel, B. P., Walker, A. R., & Barril, C. (2019). Grape berry flavonoids: a review of their biochemical responses to high and extreme high temperatures. Journal of experimental botany, 70(2), 397–423. https://doi.org/10.1093/jxb/ery392

Kliewer, W. M., & Torres, R. E. (1972). Effect of Controlled Day and Night Temperatures on Grape Coloration. American Journal of Enology and Viticulture, 23(2), 71–77. https://doi.org/10.5344/ajev.1972.23.2.71

Hochberg, U.; Batushansky, A.; Degu, A.; Rachmilevitch, S.; Fait, A. Metabolic and Physiological Responses of Shiraz and Cabernet Sauvignon (Vitis vinifera L.) to Near Optimal Temperatures of 25 and 35℃. Int. J. Mol. Sci. 2015, 16, 24276-24294. https://doi.org/10.3390/ijms161024276

Reviewer 2 Report

Comments and Suggestions for Authors

General comments to the authors

I have thoroughly reviewed the Manuscript entitledCombined metabolome and transcriptome analysis revealed the accumulation of anthocyanins in grape berry under high temperature stress”. The study highlights that high temperatures reduce the accumulation of anthocyanins in two grape varieties. Authors have examined the regulatory mechanism of anthocyanin biosynthesis under high-temperature environments. They studied  the effects of high temperature stress on berry coloration and anthocyanin biosynthesis. Results clearly demonstrated that when grapes were subjected to high temperature stress the total anthocyanin conent was decreased significantly as compared to control, while the content of procyanidins increased significantly. Anthocyanin-targeted metabonomics identified eight different types of anthocyanins including cyanidins, delphinidins, malvidins, pelargonidins, peonidins, petunidins, procyanidins, and flavonoids. Malvidins were the most abundant in the two grape varieties, with malvidin-3-O-glucoside being more sensitive to high temperatures. Also high temperature downregulated the expression of structural genes and regulators involved in the anthocyanin synthesis pathways. They used the WGCNA method to identify two modules correlated with total anthocyanin and procyanidin contents. They found that MYBCS1, bHLH137, WRKY65, WRKY75, MYB113-like, bZIP44, and GST3 were predicted to be involved in grape anthocyanin biosynthesis.

Specific comments to the authors

1. The authors are able to draw strong aims and objectives.

2. Please mention the scientific name of the grape.

3. In abstract the gist of salient findings may be given quantitatively instead of qualitative statements.

4. Please mention the full form of WGCNA in abstract.

5. Modify the keywords. I suggest the authors do not repeat the title words in key words.

6. Introduction is well written and the study is justified in view of literature.

7. Methodology is written well.

8. In the materials and methods year of experiment is not motioned whether the study is conducted in one year. Why the study was not repeated.

9. How the temperature in polyhouses was maintained?.

  10. There are various grammatical errors throughout the manuscript. I recommend the authors to rewrite without any grammatical errors.

11. I suggest the authors to add the LC-MS profiles of different grape varieties

12. In which way, the authors were trying to claim the novelty of the proposed work? There are several research papers has been published earlier. Hence, I suggest the authors to add few sentences about the novelty statement about your work.

13. KEGG pathway enrichment analysis of different metabolites is difficult to read and looks a bit blurry.

13. The discussion mainly reiterates the results while also containing simplistic interpretations. Discussion part should be improved.

14. References are adequate. The formats of the references are not uniform, you should check them carefully. Please replace old references with new references.

 15. The conclusions needs more work; you should mention the implications as well. It should contain conclusions of significance not merely vague statements.

16. Final Note I can advise the publication of the manuscript after revision.

Author Response

Point 1: The authors are able to draw strong aims and objectives.

Response 1: Thanks for your recognition.

Point 2: Please mention the scientific name of the grape.

Response 2:We added the scientific name of the grape

Point 3: In abstract the gist of salient findings may be given quantitatively instead of qualitative statements.

Response 3: We have made modifications to the abstract.

Point 4: Please mention the full form of WGCNA in abstract.

Response 4: The full form of WGCNA in abstract has been indicated.

Point 5: Modify the keywords. I suggest the authors do not repeat the title words in key words.

Response 5: We have changed some keywords

Point 6: Introduction is well written and the study is justified in view of literature.

Response 6: Thanks for your recognition.

Point 7: Methodology is written well.

Response 7: Thanks for your recognition.

Point 8: In the materials and methods year of experiment is not motioned whether the study is conducted in one year. Why the study was not repeated.

Response 8: This experiment was completed in 2020. Each plot has 15 well growing and healthy XH and FL grape plants. And set up three biological replicates, each with 5 grape plants.

Point 9: How the temperature in polyhouses was maintained?.

Response 9: The experimental sites for grape cultivation under control and high-temperature treatment are two independent communities. The indoor temperature is strictly monitored by temperature sensors, mainly using ventilation fans and controlling the size and ventilation time of greenhouse vents to regulate the temperature. In addition, we laid black weed proof cloth between the rows of grape plants, effectively suppressing weed growth, and sprayed water on the ground to reduce ground temperature and promote air circulation. During periods of extreme heat, we also use shade nets to cool down to maintain the design temperature.

Point 10:There are various grammatical errors throughout the manuscript. I recommend the authors to rewrite without any grammatical errors.

Response 10: We have corrected the grammar issue.

Point 11:I suggest the authors to add the LC-MS profiles of different grape varieties

Response 11: We have added the LC-MS profiles of different grape varieties in Figure S2.

Point 12: In which way, the authors were trying to claim the novelty of the proposed work? There are several research papers has been published earlier. Hence, I suggest the authors to add few sentences about the novelty statement about your work.

Response 12: In the introduction, we describe the innovations of this experiment.

Point 13: KEGG pathway enrichment analysis of different metabolites is difficult to read and looks a bit blurry.

Response 13: We have adjusted the image clarity.

Point 14: The discussion mainly reiterates the results while also containing simplistic interpretations. Discussion part should be improved.

Response 14: We have conducted a more in-depth analysis of the discussion section.

Point 15: References are adequate. The formats of the references are not uniform, you should check them carefully. Please replace old references with new references.

Response 15: We have standardized the format of the references.

 Point 16: The conclusions needs more work; you should mention the implications as well. It should contain conclusions of significance not merely vague statements.

Response 16: We have made modifications to the conclusion section.

Point 17: Final Note I can advise the publication of the manuscript after revision.

Response 17: Thanks for your recognition.

Reviewer 3 Report

Comments and Suggestions for Authors

Some graphs would benefit from larger fonts and clearer legends to improve readability.

Why was 35 degrees chosen as the control temperature? What is the average temperature during fruit ripening in China, and what is the optimal temperature for grape cultivation?

Line 105-106: Does this temperature refer to the whole growing period or specifically to berry development? Line 108 mentions that it is 2.35 ℃ higher than in CK. Please clarify.

How was the temperature controlled inside the heat stress greenhouse, especially considering the significant temperature decline between August 11-18 and June 23-30?

How did you separate the peel and pulp? Was it done manually?

I believe Figure 1A is not necessary.

What was the experimental design?

It would be helpful to introduce the labeling in the methods section or at the beginning of the results section. I was confused at the start of the results section until I found that XCK, XT, FCK, and FT represent the control and treatment groups of XH and HT, as explained under Figure 2 legend.

I suggest rewriting the results for clarity. For example, this sentence is difficult to follow: "The CIRG of FL showed a trend of first increasing and then decreasing, with FCK higher than FT from S1 to S3, reaching the maximum in S3, and FT higher than FCK at S4."

The conclusion can be shorter and should focus on the main findings of the study.

Author Response

Point 1: Why was 35 degrees chosen as the control temperature? What is the average temperature during fruit ripening in China, and what is the optimal temperature for grape cultivation?

Response 1: Grape varieties are highly dependent on high temperatures (HT), and vines grown in HT for a long time may adapt to increased temperatures, and their optimum temperature can vary with changes in ambient temperature. And with global warming, more and more studies are revealing how grapes respond to temperature changes in the optimal range (25-35℃). The results show that the optimal growing temperature for the vines depends on the variety. Some heat-tolerant varieties are more sensitive to temperatures of 35℃ and may show a stronger response to the higher temperatures predicted in the near future. And even if just attempting to compare effects of day and night temperatures on flavonoid metabolism in grape berries, means and/or maxima ranging from 20℃ to 45℃ have been applied at night and from 30℃ to 49 ℃ during the day. Due to the special geographical location of Xinjiang, most areas have experienced high temperatures of 35-40℃ in summer in recent years. There are even short-term extreme high temperature days of more than 40°C, and the average number of high temperature days in Xinjiang in summer can reach 25 days. Due to the lack of timely ventilation and insufficient ventilation, the grapes cultivated in the protected area were even subjected to high temperature stress of more than 45℃ in the middle of summer, which affected the growth and development of grapes and seriously affected the quality and yield of grape fruits. In order to better adapt the actual production environment of the grapes, we chose 35°C as the control, which may still be higher than other regions, but it is a common temperature for Xinjiang. And 40°C as the treatment group, which is more in line with the extreme heat in Xinjiang. Better evaluation of the actual adaptability of grapes to high temperature stress.

Point 2: Line 105-106: Does this temperature refer to the whole growing period or specifically to berry development? Line 108 mentions that it is 2.35℃ higher than in CK. Please clarify.

Response 2: As shown in Figure 1B, the growing period of grape berries is concentrated from early June to the end of August, so this temperature specifically refered to berry development. The average daily temperature of the high-temperature treatment group was 2.35°C higher than that of the control group during the whole berry development period.

Point 3: How was the temperature controlled inside the heat stress greenhouse, especially considering the significant temperature decline between August 11-18 and June 23-30?

Response 3: The experimental sites for grape cultivation under control and high-temperature treatment are two independent communities. The indoor temperature is strictly monitored by temperature sensors, mainly using ventilation fans and controlling the size and ventilation time of greenhouse vents to regulate the temperature. In addition, we laid black weed proof cloth between the rows of grape plants, effectively suppressing weed growth, and sprayed water on the ground to reduce ground temperature and promote air circulation. During periods of extreme heat, we also use shade nets to cool down to maintain the design temperature.

The significant drop in temperature during the test was caused by rainy weather. While greenhouse experiments are not without human intervention, they are closer to real field conditions and can be studied under near-normal production conditions. However, on-site manipulation is not easy to control and is more suitable for providing a temperature difference from the environment rather than absolute temperature control. While caution is needed when transferring conclusions from greenhouses to fields, the observed trends are generally consistent with field experiments.

Point 4: How did you separate the peel and pulp? Was it done manually?

Response 4: Randomly selected 50 grapes from the top, middle, and bottom of each grape cluster. The grapes were cut open with tweezers on the workbench and quickly separated the skins from the pulp, and the pulp tissue was further removed on the inside of the skins with tweezers. Then, the peel and pulp were rapidly frozen in liquid nitrogen and stored at -80°C.

Point 5: I believe Figure 1A is not necessary.

Response 5: After careful consideration, we agree with your viewpoint and have removed Figure 1A.

Point 6: What was the experimental design?

Response 6: The unique day-night temperature difference in Xinjiang is conducive to the accumulation of more sugar in grape berries, and longer light exposure time is also beneficial for fruit coloring, resulting in better fruit quality than other regions with smaller temperature differences. And the color of the fruit can more directly attract consumers' attention, thereby generating greater economic benefits. On the other hand, berries rich in anthocyanins also have greater advantages in antioxidant and anti-cancer properties. However, in recent years, the impact of extreme heat weather has had a direct impact on grape yield, while also having an inhibitory effect on fruit coloring. Therefore, whether it is breeding varieties with strong heat resistance or improving the effect of high temperature on fruits through plant regulators, greenhouse regulation, and other means, it is necessary to first clarify the molecular mechanism of high temperature on anthocyanin synthesis as a theoretical basis. Therefore, we designed the following experiment:

This experiment was conducted in 2020 in a greenhouse of the experiment station at Shihezi University, located in Shihezi, Xinjiang, China (86°4′E, 44°18′N). The plant materials used were ‘Summer Black’ (XH, a hybrid of Vitis vinifera× Vitis labrusca from Japan) and ‘Flame Seedless’ (FL, Vitis vinifera L, originating from the United States). Both varieties were grown in cultivation bags measuring 27 cm (height) × 30 cm (diameter), using a culture medium composed of pastoral soil and organic matter in a ratio of 2:1. The plant row spacing was set at 100 cm × 120 cm, and soil moisture levels was maintained between 31% ~ 35%. Each plot has 15 well growing and healthy XH and FL grape plants. And set up three biological replicates, each with 5 grape plants. The plants were pruned into a “Y” shape, with each vine having two main branches. Each of these main branches supported two berry clusters, with each fruiting branch bearing one cluster of grape bunches. Throughout the entire experimental process, all other agronomic operations were consistent.

To simulate a natural high-temperature environment, our experiments were conducted in two independent solar greenhouses. The control group (CK) was set to 35℃ ± 2℃, and the high-temperature group (HT) group was set to 40℃ ± 2 ℃. The indoor temperature was strictly monitored by temperature sensors, mainly using ventilation fans and controlling the size and ventilation time of greenhouse vents to regulate the temperature. In addition, we laid black weed proof cloth between the rows of grape plants, effectively suppressing weed growth, and sprayed water on the ground to reduce ground temperature and promote air circulation. During periods of extreme heat, we also used shade nets to cool down to maintain the design temperature. During the berry development stage (June to August), the average day/night temperature was 32.54/ 20.74℃ in the HT treatment group and 30.19/ 20.49℃ in the CK treatment group. The day/night temperature difference between the HT group and the CK group was (ΔT=2.35℃/ 0.24℃) (Figure 1A). The HT period was from 10am to 8pm daily, and temperature management was consistent for other periods (Figure 1B). At the same time, we useed the MicroLite-U disk temperature recorder to strictly monitor greenhouse temperature.

The first sample was collected at the beginning of berry coloring (about 5% coloration of whole panicle). Then, every 10 days, samples were taken until the grape berries were completely colored. Thirty grape clusters were randomly selected for each grape varieties. A total of 100 grape berries were randomly picked from the top, middle and bottom of grape clusters, and immediately packed in an ice box and transported to laboratory. The peel and pulp were separated. The peels were quickly frozen in liquid nitrogen and stored at -80℃ for further processing. Some fresh berries were used to determine the color index of red grape (CIRG).

Point 7: It would be helpful to introduce the labeling in the methods section or at the beginning of the results section. I was confused at the start of the results section until I found that XCK, XT, FCK, and FT represent the control and treatment groups of XH and HT, as explained under Figure 2 legend.

Response 7: Based on your suggestion, we acknowledge your viewpoint and introduce explanations for the abbreviations XCK, XT, FCK, and FT at the beginning of the results section.

Point 8: I suggest rewriting the results for clarity. For example, this sentence is difficult to follow: "The CIRG of FL showed a trend of first increasing and then decreasing, with FCK higher than FT from S1 to S3, reaching the maximum in S3, and FT higher than FCK at S4."

Response 8: We have checked and modified the results section, for example

In terms of appearance, with the growth and development of grape berries, XH and FL grapes had poorer berry coloring compared to the control berry in the first three stages (S1, S2, S3). However, when the berry ripens (S4), there was no significant difference (Fig 2A). Based on the CIRG index, the berry of FL was classified as dark red (5 < CIRG < 6) and XH was classified as blue-black (CIRG > 6.0) at S4 (Fig 2B and 2C). The CIRG of FL showed a trend of first increasing and then decreasing. In the first three stages of FL grape berry development (S1 to S3), the CIRG of the control group (FCK) was higher than that of the HT treatment group (FT), and the maximum value reached at S3. Only in the S4 phase, FCK's CIRG was lower than FT. In addition, the berry color of FCK reached dark red in S2, while the berry color of FT reached dark red in S3 (Fig 2C). The CIRG of XH grape berries showed an overall upward trend. The CIRG of the XH grape control group (XCK) was consistently higher than that of the HT treatment group (XT) of XH grapes. And the berry color of XCK reached dark red in the S3 stage, and reached blue-black in the S4 stage. While the berry color of XT reached dark red in the S4 stage. (Fig 2B).

Point 9: The conclusion can be shorter and should focus on the main findings of the study.

Response 9: We have provided a more precise description of the conclusion section, focusing more on the main findings of the study.

Point 10: Are the methods adequately described?

Response 10: We provided a supplementary description of the study methods.

Point 11: Are the results clearly presented?

Response 11: We complement the analysis of the results.

Point 12: Are the conclusions supported by the results?

Response 12: We rewrite the conclusions.
